Optimizing maize germination forecasts with random forest and data fusion techniques

Wu Lili 1 wulili@henau.edu.cn
Xing Yuqing 2
Yang Kaiwen 1
Li Wenqiang 1
Ren Guangyue 2
Zhang Debang 3
Fan Huiping 4 fanhuiping1972@hotmail.com
1 College of Sciences, Henan Agricultural University , Zhengzhou , China
2 College of Food and Bioengineering, Henan University of Science and Technology , Luoyang , China
3 Zhengzhou Wangu Machinery Co., Ltd , Zhengzhou , China
4 College of Food Science and Technology, Henan Agricultural University , Zhengzhou , China
Asif Muhammad
Electronic publication date: 2024 Nov 28
Publication date: 2024
Volume: 10
Electronic Location ID: e2468
Received 2024 Aug 8; Accepted 2024 Oct 9
Copyright: © 2024 Wu et al.
Copyright year: 2024
Copyright holder: Wu et al.
License: This is an open access article distributed under the terms of the Creative Commons Attribution License, which permits unrestricted use, distribution, reproduction and adaptation in any medium and for any purpose provided that it is properly attributed. For attribution, the original author(s), title, publication source (PeerJ Computer Science) and either DOI or URL of the article must be cited.
License URL: https://creativecommons.org/licenses/by/4.0/

Keywords: Maize seeds, Germination rate, Non-destructive prediction, Random forest algorithm, Image processing

Funding: Major science and technology project of Henan Province 221100110800 This work was supported by the Major science and technology project of Henan Province (grant numbers 221100110800). The funders had no role in study design, data collection and analysis, decision to publish, or preparation of the manuscript.

==============================
Traditional methods for detecting seed germination rates often involve lengthy experiments that result in damaged seeds. This study selected the Zheng Dan-958 maize variety to predict germination rates using multi-source information fusion and a random forest (RF) algorithm. Images of the seeds and internal cracks were captured with a digital camera. In contrast, the dielectric constant of the seeds was measured using a flat capacitor and converted into voltage readings. Features such as color, shape, texture, crack count, and normalized voltage were used to form feature vectors. Various prediction algorithms, including random forest (RF), radial basis function (RBF), neural networks (NNs), support vector machine (SVM), and extreme learning machine (ELM), were developed and tested against standard germination experiments. The RF model stood out, with a training time of 5.18 s and the highest accuracy of 92.88%, along with a mean absolute error (MAE) of 0.913 and a root mean square error (RMSE) of 1.163. The study concluded that the RF model, combined with multi-source information fusion, offers a feasible and nondestructive method for quickly and accurately predicting maize seed germination rates.

Introduction

China produces maize, one of the country’s significant crops, which has substantially impacted grain production and farmers’ income. In maize cultivation, high-vitality seeds greatly impact the maize’s quality and yield amount. To achieve a sound increase in maize yield, it is necessary to ensure that the seeds have good vitality. The germination rate of seeds is a significant indicator of seed vitality. It is a fact that seed vitality and germination rate are positively associated, and germination rate is one of the substantial indicators to assess seed quality.

To ensure a high yield, manual germination tests are conducted before sowing to find the germination rate of seeds (Zhang et al., 2024). The conventional methods for detecting seed vitality are seedling growth tests, tetrazolium staining methods, cold resistance tests, accelerated aging tests (Hao et al., 2023), and electrical conductivity tests (Marin et al., 2018). However, these methodologies have deficiencies such as high labor and time costs, damaged seeds due to destructive testing methods, and being affected easily by environmental conditions. To address these problems, an alternative method of staining was suggested. Nevertheless, it is also time-consuming and affected by the seed dormancy period, is not helpful for rapid detection, and requires professional operators, thus leading to concerns regarding the reliability and stability of test outcomes. Hence, computational methods combined with chemometrics methodologies that can predict the germination rate quickly and nondestructively need to be developed is an urgent issue (Hao et al., 2023; Xie et al., 2023; Zhang et al., 2020; Yi et al., 2022).

The advancements in artificial intelligence (AI), machine vision (MV), and spectral imaging technology (SIT) have extensively promoted the progress of nondestructive testing technology for agricultural products. The nondestructive testing methods for seed vitality (germination rate) used mainly include near-infrared spectroscopy (NIS), hyperspectral imaging (HS-I), and machine vision (MV).

When computational methods are implemented to find the germination rate of seeds, two groups appear to be influential and utilize state-of-the-art data collection approaches, namely, hyperspectral images. While the first group consists of machine-learning approaches, the second group comprises AI-based algorithms (Jiang et al., 2024; Hao et al., 2023; Bai et al., 2021; Javanmardi et al., 2021; Liu et al., 2024; Nucci et al., 2023; Song et al., 2022; Tu et al., 2021; Wang et al., 2023).

Yang et al. (2021) employed near-infrared spectroscopy (NIS) and MV methods to detect seed vitality. NIR was used to measure maize seeds’ spectrum information, and the vitality detection model was established using the K-nearest neighbor (KNN) method. The accuracy and recall rates and F1 score were 86.4%, 71.9%, and 78.5%, respectively. The maize seed images were captured through an image acquisition platform, and eight principal components of features were extracted using principal component analysis (PCA). The corn seed vitality detection model was constructed using the extreme learning machine (ELM) method. The outcomes indicated that the ELM's precision, recall rate, and F1 score for seed vitality were 89.0%, 66.8%, and 76.3%, respectively.

Xu et al. (2022) suggested a method for identifying maize seeds’ vitality using hyperspectral imaging and multivariate data analysis. The Zhengdan 958 varieties with 1,680 maize seeds were subjected to artificial aging treatment. The vitality differences between damaged and healthy seeds were verified by utilizing the outcomes of the standard germination test. The HS-I gathered the samples’ spectral measurements, and different machine recognition algorithms were constructed by employing full wavelength and feature wavelength. The experimental results showed that DE-UVE-ANN (detrending-uninformative variable elimination-artificial neural network) was the best algorithm with a 95.24% accuracy. The two other algorithms, linear discriminant analysis (LDA) and artificial neural networks (ANNs), had accuracy of 85.71% and 89.76%, respectively.

Huang et al. (2019) proposed the 3-vigor gradient classification algorithm employing machine learning and deep learning algorithms combined with HS-I for maize seeds. The results showed that the principal component analysis-support vector machine (PCA-SVM) algorithm achieved the best result of pure spectral features with an accuracy of 92.5% in the test set. The SVM reached the best outcomes of fusion features with an accuracy of 93.1% in the test set. The lightweight Mobile Net reached the maximum accuracy of 99.5% in the test set.

Wang et al. (2022) constructed a fast, nondestructive, and efficient classification algorithm for the vitality levels of maize seeds by utilizing infrared thermal imaging technology combined with the SVM algorithm. The infrared thermal image of maize seeds was collected through an infrared thermal imager, temperature scores were extracted as features, and SVM was implemented for training. The outcomes designated that the constructed algorithm using infrared thermal imaging technology combined with the SVM has an accuracy of 92.4% and 91.0% in the training and test sets, respectively, with 0.12 s training duration. After the optimization was run, the model’s accuracy reached 97.1% and 96.5% in the training and the test sets, respectively.

Liu, Chen & Jiao (2023) suggested an algorithm that rapidly detects the germination rate of maize seeds by utilizing an enhanced local linear embedding and NI-S. The results designated that the cosine similarity can represent the spectral data of aged maize seeds. The SVM, a nono-linear model, fits better to predict maize seeds’ germination rate when compared with PLS.

Xiao et al. (2023) proposed an algorithm that quickly detects the germination rates of maize seeds that employ the Gaussian regression model by implementing NI-S. Two hundred forty-five maize seeds with three distinct varieties are artificially aged to obtain varying degrees of vitality.

The NIS of each sample was collected, and the different germination rate prediction models were constructed by the PLS, SVM, and Gaussian Process Regression (GPR) utilizing distinct kernel functions. The experiment designated that the Matern32-GPR model detected the germination rates of maize seeds with greater stability and superior capability.

Zhang et al. (2023) combined HS-I and non-targeted metabolomics by employing PLS-R, SVM-R, and OPLS-DA to estimate the loss of vitality of naturally aged sweetcorn seeds. Xu et al. (2022) suggested a nondestructive detection algorithm for maize seed vitality by implementing visible/near-infrared spatially resolved spectroscopy (SRS) unified with chemometrics. The maximum precision for both S1 (the embryonic side) and S2 (endosperm side) in the Zhengdan-958 variety was 91.67%, while those of S1 and S2 for the Shaandan-650 variety were 86.67% and 88.33%, respectively. Moreover, the SRS was found more advantageous in S2 acquisition, validating the SRS’s potential when seed vitality is tested nondestructively.

A single detection technique can only reflect information about a certain aspect of a target. A fusion of multi-source information increases the dimensionality and confidence of detection, expands the observation range of time and space, and can more comprehensively reflect the information of a target from distinct perspectives. So it can improve the accuracy of detection.

The research intends to integrate information from two distinct sources, capacitive sensors and machine vision, to perform germination rate nondestructively on maize seeds. The random forest (RF) algorithm will be employed to predict the germination rate of maize seeds based on fused datasets, which will provide a reference and basis to test the quality of maize seeds rapidly and nondestructively. Thus, the research offers 2-fold benefits for investigating the germinate rate of seeds, namely, novel data collection and its usage in the modeling phase and an effective and practical model called the RF.

Materials and Methods

Materials

The Zhengdan-958 maize seed variety utilized experimentally was purchased from Zhengzhou, Henan Province, in 2023. The germination rate of the Zhengdan-958 variety was about 91%. The 600 undamaged seeds were randomly chosen as samples to experiment.

Image acquisition

The maize seed images were obtained using two different methods: top lighting to get the appearance image and bottom lighting to attain the internal crack image.

Appearance image acquisition

The sample images of maize seeds were captured using a PSSX60HS digital camera. The acquisition of maize appearance images was completed in a lighting room to avoid the influence of external light. The diffuse reflective circular LED light was utilized as the light source for indoor illumination. A black, non-reflective paper was chosen as a background to discriminate maize seeds from the background. The light source was placed above the maize, and the seeds with the embryo facing upwards were placed sequentially on black paper.

The 600 seeds are randomly distributed in 12 groups, with 50 maize seeds in each image. After smoothing, the image was uniformly cropped to 1,130 × 700 pixels, as shown in Fig. 1, which was the appearance image of the first group of maize seeds.

Figure 1 Maize seed appearance image.

Internal crack image acquisition

The internal crack image collection of maize was also completed in a lighting room. The seeds with the embryo facing downwards were placed sequentially on a white paper. Internal cracks inside seeds were difficult to identify and needed to be exposed to light source illumination (Li et al., 2024). Therefore, the light sources were placed under a white paper. Cracked seeds promoted germination (46%) with a mean germination time of 146 days (Iralu & Upadhaya, 2018). Like the process of capturing appearance images, internal crack images were also composed of a 50-seed image. Figure 2 shows the internal crack image of the first maize group.

Figure 2 Maize seed internal crack image.

Dielectric constant—voltage score acquisition

The dielectric constant of a maize seed can be employed to characterize a seed’s internal quality, and differences in seed moisture content, thickness, protein, fatty acid content, etc. can all affect the dielectric constant. Two electrically connected metal plates are employed to form a parallel plate capacitance sensor, and the output capacitance score will change with alterations in the plate area, plate spacing, or the medium between the plates. When the area and spacing of the plates remain unchanged, but the dielectric constant between the plates changes, the dielectric constant will change accordingly, thereby altering the capacitance score. By converting the amplification circuit, capacitance alterations are converted into voltage changes.

Each maize seed was placed between two flat electrodes to obtain the corresponding voltage score. Figure 3 shows the normalized voltage scores of the first group of maize seeds between the flat electrodes.

Figure 3 The normalized voltage values of the first group of maize seeds.

Germination experiment

Before running a standard germination test on Zhengdan 958 variety seeds, the seeds were soaked in a germination dish disinfected with alcohol for 12 h. Then, the seeds were covered with sprouting paper and cling film, and the seeds were watered regularly every day to replenish moisture, allowing them to grow in a moist environment. The incubator temperature was 25 °C, and the germination dishes were placed in it. The process of seed germination is shown in Fig. 4.

Figure 4 Maize seeds germination process.

The germinated seeds on the 7th day were recorded utilizing the samples’ germination and rooting criteria. A consistent germination rate of 91% was found, with the same rate in large samples. Therefore, a certain representation for the 600 samples was shown.

Data analysis and processing

The segmentation of a maize image

To facilitate the feature extraction of single corn seeds, the background of each maize needs to be blackened. So, maize images can be better seen when the background turns black. In other words, the target area of the maize seed had a large grey difference from the background. Then, the histogram method was implemented to pick the optimal threshold, and the segmentation outcome is presented in Fig. 5.

Figure 5 Background segmentation of maize seed appearance image.

In the maize internal crack images, the background was white, and there was a slight difference in color between the maize and the background. So, the segmentation method based on grayscale histograms was not very suitable (Xu, Li & Chen, 2022). The commonly used image segmentation algorithms include Sobel’s edge detection, Prewitt’s edge detection, Roberts’s edge detection, and Canny’s edge detection algorithms. A background segmentation needs to separate individual maize grains and reflect the crack information inside the maize. Figure 6 shows the maize images obtained by implementing distinct edge detection methods, and the segmentation impact is not ideal.

Figure 6 Segmentation effects of different edge operators.

(A) Original image, (B) preprocessed image, (C) grayscale image, (D) Sobel edge detection, (E) Prewitt edge detection, (F) Roberts edge detection, (G) LoG edge detection, (H) canny edge detection.

A maize crack image was first processed by mathematical morphology and Sobel operator and then refused through multi-scale decomposition of the wavelet transform. The new image is shown in Fig. 7. The edges of the image are clear, complete, and continuous. The image not only retains internal crack information but also removes unnecessary noise (Qiu et al., 2022).

Figure 7 Segmentation effect after improved algorithm.

The feature extraction of the appearance image

Color feature

Color attributes are the most intuitive and obvious physical variables of an image. Compared to geometric features, color features have a certain degree of stability and are insensitive to size and direction. The images captured by the camera are RGB representations, and the HSI model is consistent with the human eye’s observation mode. Therefore, the RGB and HSI representations were employed in the manuscript to extract color features.

In RGB, representations are superimposed to obtain colors that can be perceived visually. Due to the fusion of red and green resulting in yellow, R and G components were selected for maize color feature extraction.

In the HIS representation, the I component is independent of color information. The light source and intensity that affect the S component remain unchanged in actual shooting, so the S’s component score is determined. Thus, only the H component must be considered in maize color recognition.

R, G, and H components were derived from the preprocessed sample images and normalized for subsequent model training. Figure 8 shows the color characteristic scores of the first group of 50 maize seeds.

Figure 8 The color characteristics of the first group of maize seeds.

Shape feature

(a) Length-width ratio

Each maize seed was divided and a bounding rectangle’s length and width were employed to represent the length and width of a seed, as shown in Fig. 9.

Figure 9 Labeled image.

(b) Area and perimeter

The area of a maize is the pixels’ total number in the seed area, and the area score can be obtained by summing up the pixels’ actual numbers in the area of the maize seed. The perimeter is the total number of pixels at the boundary of the maize seed area.

(c) Rectangularity and circularity

Rectangularity represents the degree to which a corn seed is close to a rectangle. Equation (1) is used to calculate rectangularity.

(1) Re=ALW

where Re, A, L, and W represent the rectangle, seed area, seed length, and seed width, respectively.

Circularity represents the degree to which the shape of a corn seed is close to a circle. Equation (2) is utilized to compute the circularity score.

(2) C=4πAP2

where C, A, and P represent the circularity, the area of the seed, and the circumference of the seed, respectively.

(d) Embryo area ratio

There are apparent indentations on the embryonic surface of corn seeds, which is the boundary between the embryonic area (white part) and the nonembryonic area (yellow part), as shown in Fig. 10. The ratio of embryo bud area can be utilized to distinguish maize seeds. Equation (3) is employed to calculate the embryo area ratio.

(3) E=EmA

where E, A, and Em represent the embryo area, the area of the seed, and the area of the embryo bud, respectively.

Figure 10 Schematic diagram of embryo area ratio.

Figure 11 illustrates the six shape features extracted from 50 seeds of the first maize samples.

Figure 11 The shape characteristics of the first group of maize seeds.

Texture feature

The texture is locally irregular and is composed of macroscopic characteristics in an image, which is a natural attribute of the surface of an object and one of the commonly utilized image features. The methods extracting texture features mainly include structural, statistical, and spectral approaches. The grey level co-occurrence matrix (GLCM), a broadly implemented statistical method, is employed to analyze image texture features proposed by Haralick.

The GLCM was implemented to extract texture features. The energy, contrast, correlation, and homogeneity of the image were computed from four directions, namely, 0°, 45°, 90°, and 135° to represent the texture features. The expressions are shown in Eqs. (4) to (7).

(a) Energy

(4) ENR=∑i=0L−1∑j=0L−1P(i,j|d,θ)2

(b) Contrast

(5) CON=∑i=0L−1∑j=0L−1(i−j)2P(i,j|d,θ)

(c) Correlation

(6) COR=∑i=0L−1∑j=0L−1ijP(i,j|d,θ)−μ1μ2σ1σ2

where μ1,μ2,σ1,σ2 are expressed by

μ1=∑i=0L−1i∑j=0L−1P(i,j|d,θ), μ2=∑i=0L−1j∑j=0L−1P(i,j|d,θ)

σ1=∑i=0L−1(i−μ1)2∑j=0L−1P(i,j|d,θ), σ2=∑i=0L−1(j−μ2)2∑j=0L−1P(i,j|d,θ)

(d) Homogeneity

(7) HOM=∑i=0L−1∑j=0L−1P(i,j|d,θ)1+(i−j)2.

From Eqs. (4) to (7), P(i,j|d,θ) represent the probabilities of two pixels i and j in the image with a distance of d, and θ represents the four directions of 0°, 45°, 90°, and 135°, respectively.

As mentioned above, the same feature extraction process was performed on all maize seed samples. Figure 12 shows the texture features of 50 maize seeds used as training samples in the first group.

Figure 12 The texture characteristics of the first group of maize seeds.

Internal crack feature

Single-grain maize needed to be segmented, and the number of internal cracks also needed to be counted for image processing of internal cracks. According to the number of cracks in a maize seed, the one containing one crack, two cracks, and three or more cracks were called a single crack, a double, and a turtle crack, respectively.

Each crack had only one starting and ending point for single and double-cracked maize. The starting and ending points of the cracks are found to compute the crack numbers. There were at least turtle cracks or many single cracks for turtle-cracked maize. For statistical convenience, the branch of the turtle crack was also treated as a crack. To search for the two-dimensional logical matrix of a maize crack image, the number of cracks could be counted. Figure 13 shows the first class's internal crack numbers in 50 corn seeds.

Figure 13 The internal cracks number of the first group of maize seeds.

Multi-source feature fusion

The multi-source heterogeneous data can be fused by dividing it into layers, such as the data, feature, and decision layers. The fusion of the feature layer utilizes a greater amount of information and results in higher accuracy when compared to the fusion of the decision layer. Compared to data layer fusion, it eliminates redundant information and reduces computational complexity. Therefore, the fusion of the feature layers was chosen in the experiment. The data fusion method based on multi-core learning can flexibly employ distinct kernel functions to process data from different sources and has been widely applied in classification and prediction tasks. The weighted polynomial extension was implemented to fuse the extracted features mentioned above.

Random forest regression

The RF employs an ensemble learning proposed by Beriman (2001) to reduce overfitting risk and improve overall model performance by constructing multiple decision trees. The RF employs the bootstrap sampling method to randomly generate multiple training sets and produce corresponding decision trees for each training set. Each decision tree is predicted once, and the average of the predicted scores from numerous decision trees is combined to generate the result. The RF adds additional randomness to the model while growing the trees. Instead of searching for the most critical feature while splitting a node, it searches for the best feature among a random subset of features. This results in a wide diversity that generally results in a better model.

Figure 14 depicts the schema of the RF.

Figure 14 Random forest algorithm flow diagram.

(a) A training set is generated for each decision tree by randomly selecting data from the original dataset with no duplicates or replacements.

(b) Each training set could be implemented to construct a decision tree where no pruning is required.

(c) An RF was composed of multiple decision trees. The RF’s prediction is the mean of the results of all decision trees. Also, the steps of the RF are presented as follows:

Step 1: Pick random samples from a given dataset.

Step 2: Construct a decision tree for each training data.

Step 3: Vote by averaging the decision tree.

Step 4: Eventually, pick the most voted estimation outcome as the outcome.

The numbers of decision trees and leaves represent the key parameters affecting the RF model’s predictive capability.

Evaluation index

MAE and RMSE are implemented to assess the RF model’s prediction accuracy (Özen, 2024). The closer the MAE and RMSE scores near 0, the higher the model’s accuracy. The scores of MAE and RMSE can be calculated by Eqs. (8) and (9).

(8) MAE=1N∑i=1N|yi−y^i|

(9) RMSE=1N∑i=1N(yi−y^i)2.

The accuracy is calculated by Eq. (10),

(10) ACC=(1−∑i=1N|yi−y^i||yi|)×100%

where the number of samples is represented by N, yi and y^i the true and estimated scores are designated, respectively.

Results and discussion

The implementation of the RF prediction model

The 600 Zheng Dan-958 variety maize seeds were split into 12 classes, containing 50 in each. The split courses of 1–10 are allocated for the training and 11–12 for the testing.

In the RF regression model, two parameters must be adjusted: decision tree numbers and feature numbers when each node is split. The cross-validation is employed to determine the number of optimal parameters.

The grid search method was implemented to compute the cross-validation performance of various parameters on the new sample training set. When the numbers of the decision trees and the maximum features are assigned to 80 and 2, respectively, the minimum error would be obtained.

The predicted output scores and corresponding prediction errors of corn germination rate are shown in Fig. 15. MAE and RMSE are 0.913 and 1.163, respectively.

Figure 15 Output of random forest prediction model.

(A) Output of the prediction germination rate values, (B) output error.

The comparison results of the different strategies

When comparing the prediction results of the RF with those of the radial basis function (RBF), support vector machine (SVM), and extreme learning machine (ELM) through experiments. Table 1 presents the optimal prediction outcomes of the germination rate obtained by several distinct prediction models by implementing the same training and testing sets.

Table 1 Comparison of germination rate prediction results using different prediction models.

Prediction model	MAE	RMSE	Accuracy (%)	Elapsed time (s)	
RF	0.913	1.163	92.88	5.18	
RBF	4.726	5.392	87.57	7.49	
SVM	3.157	4.625	89.49	9.95	
ELM	1.364	1.951	90.63	4.26	

Table 1 depicts that the RF excels the RBF, SVM, and ELM when the prediction accuracy is used. The RF’s runtime is second after the ELM for model efficiency. Overall, the RF prediction model has a simple structure, easy parameter adjustment, and effectively resolves the problem of overfitting, while also improving training efficiency.

Discussion

In the research, Zheng Dan958 variety maize seeds are utilized. A digital camera and a parallel plate capacitor are employed to attain the appearance and intrinsic information of maize seeds non-destructively. To extract features and fuse distinct source information, the germination rate of maize seed is predicted by the RF with a 5.18-s running time. The accuracy reached 92.88%, the average absolute error of the prediction results was 0.913, the RMSE was 1.163. The RF provides significant advantages when compared with the RBF, SVM, and ELM. To predict the germination rate of maize, the proposed nondestructive approach employing multi-source information fusion has strong practicality and becomes a reference.

Conclusion

The Zheng Dan-958 variety was selected to predict the germination rate of maize by employing a multi-source information fusion and RF algorithm. Multi-source information comprises seed and internal crack images, the gauged dielectric constant of maize seeds using a flat capacitor converted into voltage measurements, and the features related to color, shape, texture, crack numbers, and normalized voltage scores. All are used to form feature vectors. Besides the RF algorithm, other machine learning algorithms, namely, RBF, NNs, SVM, and ELM, are used to generate predictions.

The results were compared with standard germination experiments using sampled maize seeds. The RF prediction model achieving 5.18 s of training time had the highest accuracy, reaching 92.88%. The mean absolute error (MAE) and root mean square error (RMSE) were 0.913 and 1.163, respectively. The experiment determined that the germination rate of maize seed predicted by RF nondestructively and utilizing multi-source information fusion had certain feasibility and could quickly and accurately predict the maize seed germination rate.

Supplemental Information

Supplemental Information 1 Code.

Supplemental Information 2 Raw dataset.

Additional Information and Declarations

Competing Interests

Author Contributions

Data Availability

The authors declare that they have no competing interests.

Debang Zhang is employed with Zhengzhou Wangu Machinery Co., Ltd, Zhengzhou, China.

Lili Wu conceived and designed the experiments, analyzed the data, performed the computation work, prepared figures and/or tables, and approved the final draft.

Yuqing Xing conceived and designed the experiments, analyzed the data, prepared figures and/or tables, and approved the final draft.

Kaiwen Yang conceived and designed the experiments, performed the experiments, prepared figures and/or tables, and approved the final draft.

Wenqiang Li performed the experiments, analyzed the data, performed the computation work, prepared figures and/or tables, authored or reviewed drafts of the article, and approved the final draft.

Guangyue Ren performed the experiments, analyzed the data, performed the computation work, authored or reviewed drafts of the article, and approved the final draft.

Debang Zhang conceived and designed the experiments, performed the experiments, analyzed the data, authored or reviewed drafts of the article, and approved the final draft.

Huiping Fan conceived and designed the experiments, authored or reviewed drafts of the article, and approved the final draft.

The following information was supplied regarding data availability:

The code and data are available in the Supplemental Files.

The data is available at GitHub: https://github.com/Archaeo-Programmer/cropDiffusionR.

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
