# Peer review of "Optimizing maize germination forecasts with random forest and data fusion techniques"

_PeerJ Computer Science, doi:10.7717/peerj-cs.2468_

## Round 0.1 · original submission · Major Revisions

Thank you for submitting your manuscript, After careful evaluation by our review team, we appreciate the effort and depth of research that has gone into your work. However, we as per the reviewers, the manuscript requires revisions before it can be considered for publication.

The reviewers have identified several key areas that need significant improvement, including language, and other technical issues as mentioned in their comments below. We believe addressing these concerns will strengthen the quality and impact of your research.

Please carefully review the below comments from the reviewers and make the necessary revisions. We look forward to receiving your revised manuscript.

Thank you for your understanding and cooperation.

Editor Comments:

The link between children's environment creation and behavior classification needs to be more explicitly defined
comparing the model’s performance with other state-of-the-art methods would help contextualize its success.
Some parts of the text could be written in a more concise and clear manner. Streamlining sentences and improving the flow between sections would make the paper easier to follow.


Reviewer 1 ·

Basic reporting

This study proposes the deep learning based model for predicting the germination rate to optimize the maize seed germination and use of multi-source data fusion techniques, in order to overcome the lacking abilities of conventional methodologies used for germination rate detection. Researcher used RF, RBFNN, SVM and ELM models for prediction and compared the outcomes with other conventional methods, concluded that the RF model produced the highest accuracy with non-destructive germination rate of maize seeds and feasibility of multi-source information fusion. However, this article still needs some improvements described below:
1. Language should be edited by a professional speaker and ensure that the tenses are checked properly.
2. I suggest that in the section-1 researcher should describe the research gaps and shortcomings of the existing studies and elaborate the contribution of presented study to highlight the novelty of this paper.

Experimental design

3. The research questions and objectives should be clearly stated to justify the use of presented models or methodology and also how the knowledge gaps of existing methods are addressed in presented study must be explained.
4. The author should align the contribution of this research in the abstract, introduction, and conclusion sections. Clearly articulating the unique contributions and significance of the study will help readers understand its value and relevance within the broader research domain.
5. You have provided Figure 14: Random forest algorithm flow diagram describing the use of RF algorithm for prediction of maize seed germination but manuscript would be more understandable if you add concept/architecture diagram of your proposed prediction model representing all of its phases for better understanding and contribution in future research studies.

Validity of the findings

6. Raw data and supplemental files have been provided that shows your rigorous and dedicated investigation, though the thing that weakens your manuscript is the results and discussion section that needs to be expanded. I would suggest the author to add ablation experiments/results/statistical analysis in Part 4 to prove the validity of the model.
7. Conclusion section is missing in the manuscript, as it highlights the importance of presented study, findings of existing research gaps and effectiveness or limitations of used methodology. So, I suggest to add conclusion section for better understanding and relevance.
8. The references section needs to add some good relevant articles from the latest journals.

Reviewer 2 ·

Basic reporting

The abstract provides a concise overview but should more clearly emphasize the novelty of the study, especially in terms of the specific benefits of using Random Forest (RF) and multi-source fusion compared to other models.
The description of the germination experiment is clear, but more details on the exact setup (e.g., moisture conditions, incubation) would ensure better reproducibility.
The current review tends to summarize past studies without critically analyzing their strengths, weaknesses, or gaps. A more critical assessment of how previous studies were conducted, their limitations, and how this paper addresses those gaps would strengthen the justification for the current research approach.

Experimental design

The use of both top and bottom lighting for maize seed imaging is innovative, but the paper does not clearly justify the choice of these lighting conditions over others used in prior studies. Explaining this could enhance the methodological rationale.
The method for counting internal cracks is innovative, yet the explanation of how cracks correlate with germination could be strengthened by citing additional studies that validate this relationship.
The explanation of Random Forest model parameters, such as the number of trees and feature selection, needs more depth.
The experimental setup, especially the environment for germination testing, should be described in more detail for reproducibility.

Validity of the findings

The paper discusses how voltage values are normalized for feature extraction but does not investigate into potential variances due to environmental factors like humidity, which may affect the dielectric properties of maize seeds.
The paper briefly mentions the use of RF but lacks a detailed explanation of why this model is particularly well-suited for the prediction task compared to other algorithms such as Gradient Boosting Machines.
The paper mentions the RF model's execution time but should provide insights into potential real-world applications where time efficiency is crucial, such as large-scale seed testing operations.
The section on feature extraction should include more details on how texture features were quantified and the rationale for choosing specific features.
Discuss potential confounding factors that could have influenced the results, such as variations in seed quality or environmental conditions during testing.

Additional comments

Suggest specific areas for future research, such as testing the model with other seed varieties or integrating additional data sources.

---

## Round 0.2 · Minor Revisions

Dear authors

I'm pleased to inform you about the opinion of the reviewers on your revised manuscript.

Although they are satisfied with the technical aspects of the paper. But I still feel that there are some more improvements needed in terms of language improvement, especially the abstract should be more clearer and easily understandable.

Please revise and resubmit

Reviewer 1 ·

Basic reporting

The paper has been revised well in the light of previous comments. Thus, I have no more comments.

Experimental design

no comment

Validity of the findings

no comment

Reviewer 2 ·

Basic reporting

Authors have solve all the required improvements as requested in the review.

Experimental design

Authors have solve all the required improvements as requested in the review.

Validity of the findings

Authors have solve all the required improvements as requested in the review.

---

## Round 0.3 · accepted · Accept

Dear authors

Thanks for your resubmission, I'm pleased to inform you that your manuscript is being recommended for publication. Thank you for your fine contribution to our esteemed journal.